# Analog Memories in a Balanced Rate-Based Network of E-I Neurons

**Dylan Festa**
df325@cam.ac.uk

**Guillaume Hennequin**
gjeh2@cam.ac.uk

**Máté Lengyel**
m.lengyel@eng.cam.ac.uk

Computational & Biological Learning Lab, Department of Engineering
University of Cambridge, UK

## Abstract

The persistent and graded activity often observed in cortical circuits is some-times seen as a signature of autoassociative retrieval of memories stored earlier in synaptic efficacies. However, despite decades of theoretical work on the sub-ject, the mechanisms that support the storage and retrieval of memories remain unclear. Previous proposals concerning the dynamics of memory networks have fallen short of incorporating some key physiological constraints in a unified way. Specifically, some models violate Dale's law (i.e. allow neurons to be both excita-tory and inhibitory), while some others restrict the representation of memories to a binary format, or induce recall states in which some neurons fire at rates close to saturation. We propose a novel control-theoretic framework to build function-ing attractor networks that satisfy a set of relevant physiological constraints. We directly optimize networks of excitatory and inhibitory neurons to force sets of arbitrary analog patterns to become stable fixed points of the dynamics. The re-sulting networks operate in the balanced regime, are robust to corruptions of the memory cue as well as to ongoing noise, and incidentally explain the reduction of trial-to-trial variability following stimulus onset that is ubiquitously observed in sensory and motor cortices. Our results constitute a step forward in our under-standing of the neural substrate of memory.

## 1   Introduction

Memories are thought to be encoded in the joint, persistent activity of groups of neurons. According to this view, memories are embedded via long-lasting modifications of the synaptic connections between neurons (storage) such that partial or noisy initialization of the network activity drives the collective dynamics of the neurons into the corresponding memory state (recall) [1]. Models of memory circuits following these principles abound in the theoretical neuroscience literature, but few respect some of the most fundamental properties of brain networks, including: i) the separation of neurons into distinct classes of excitatory (E) and inhibitory (I) cells – known as Dale's law –, ii) the presence of recurrent and sparse synaptic connections, iii) the possibility for each neuron to sustain graded levels of activity in different memories, iv) the firing of action potentials at reasonably low rates, and v) a dynamic balance of E and I inputs.

In the original Hopfield network [1], connectivity must be symmetrical, which violates Dale's law. Moreover, just as in much of the work following it up, memories are encoded in binary neuronal responses and so converge towards effectively binary recall states even if the recall dynamics for-mally uses graded activities [2]. Subsequent work considered non-binary pattern distributions [3, 4], and derived high theoretical capacity limits for them, but those capacities proved difficult – if not impossible – to realise in practice [5, 6], and the network dynamics therein did not explicitly model inhibitory neurons thus implicitly assuming instantaneous inhibitory feedback. More recent work

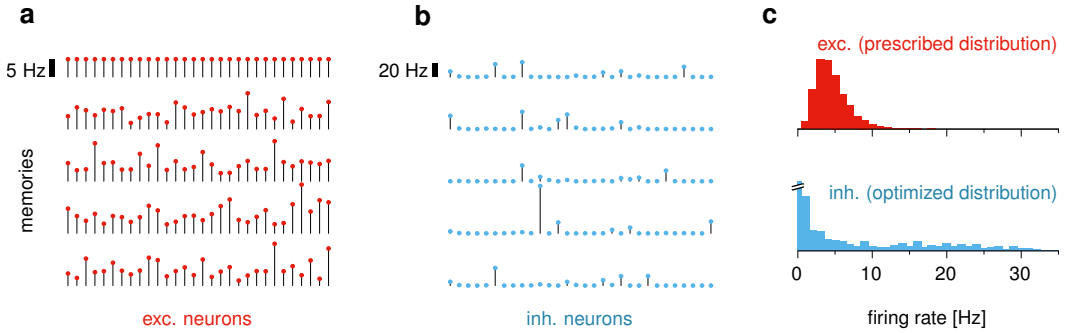

Figure 1: **(a)** Examples of analog patterns of excitatory neuronal activities, drawn from a log-normal distribution. In all our training experiments, network parameters were optimized to stabilize a set of such analog patterns and the baseline, uniform activity state (top row). For ease of visualization, only 30 of the 100 excitatory neurons are shown. **(b)** Optimized values of the inhibitory (auxiliary) neuronal firing rates for 5 of 30 learned memories (corresponding to those in panel **a**). Only 30 of the 50 auxiliary neurons are shown. **(c)** Empirical distributions of firing rates across neurons and memory patterns, for each population.

incorporated Dale's law, and described neurons using the more realistic, leaky integrate-and-fire (LIF) neuron model [7]. However, the stability of the recall states still relied critically on the saturating behavior of the LIF input-output transfer function at high rates. Although it was later shown that dynamic feedback inhibition can stabilize relatively low firing rates in subpopulations of more tightly connected neurons [8, 9], inhibitory feedback in these models is global, and calibrated for a single stereotypical level of excitation for all memories, implying effectively binary memories again. Finally, spatially connected networks are able to sustain graded activity patterns (spatial "bumps"), but make strong assumptions about the spatial structure of both the connectivity and the memory patterns, and are sensitive to ongoing noise (e.g. [10, 11]). Ref. [12] provides a rare example of spike timing-based graded memory network, but it again did not contain inhibitory units.

Here we propose a general control-theoretic framework that overcomes all of the above limitations with minimal additional assumptions. We formalize memory storage as implying two conditions: that the desired activity states be fixed points of the dynamics, and that the dynamics be stable around those fixed points. We directly optimize the network parameters, including the synaptic connectivity, to satisfy both conditions for a collection of arbitrary, graded memory patterns (Fig. 1). The fixed point condition is achieved by minimizing the time derivative of the neural activity, such that ideally it reaches zero, at each of the desired attractor states. Stability, however, is more difficult to achieve because the fixed-point constraints tend to create strong positive feedback loops in the recurrent circuitry, and direct measures of dynamical stability (eg. the spectral abscissa) do not admit efficient, gradient-based optimization. Thus, we use recently developed methods from robust control theory, namely the minimization of the Smoothed Spectral Abscissa (SSA, [13, 14]) to perform robust stability optimization. To satisfy biological constraints, we parametrize the networks that we optimize such that they have realistic firing rate dynamics and their connectivities obey Dale's law. We show that despite these constraints the resulting networks perform memory recall that is robust to noise in both the recall cue and the ongoing dynamics, and is stabilized through a tight dynamic balance of excitation and inhibition. This novel way of constructing structurally realistic memory networks should open new routes to the understanding of memory and its neural substrate.

## 2 Methods

We study a network of $n = n_\mathrm{E}$ (excitatory) $+ n_\mathrm{I}$ (inhibitory) neurons. The activity of neuron $i$ is represented by a single scalar potential $v_i$, which is converted into a firing rate $r_i$ via a threshold-quadratic gain function (e.g. [15]):

$$r_i \quad = \quad g(v_i) \quad := \quad \begin{cases} \gamma v_i^2 & \text{if} \quad v_i > 0 \\ 0 & \text{otherwise} \end{cases} \qquad (1)$$

We set $\gamma$ to 0.04, such that $g(v_i)$ spans a few tens of Hz when $v_i$ spans a few tens of mV, as experimentally observed in cortical areas (e.g. cat's V1 [16]). The instantaneous state of the system can be expressed as a vector $\mathbf{v}(t) := (v_1(t), \ldots, v_n(t))$. We denote the activity of the excitatory or inhibitory subpopulation by $\mathbf{v}_{\text{exc}}$ and $\mathbf{v}_{\text{inh}}$, respectively. The recurrent interactions between neurons are governed by a synaptic weight matrix $\mathbf{W}$, in which the sign of each element $W_{ij}$ depends on the nature (excitatory or inhibitory) of the presynaptic neuron $j$. We enforce Dale's law via a re-parameterization of the synaptic weights:

$$W_{ij} = s_j \ \log(1 + \exp \beta_{ij}) \quad \text{with} \quad s_j = \begin{cases} +1 & \text{if } j \leq n_{\text{E}} \\ -1 & \text{otherwise} \end{cases} \tag{2}$$

where the $\beta_{ij}$'s are free, unconstrained parameters. (We do not allow for autapses, i.e. we fix $W_{ii} = 0$). The network dynamics are thus given by:

$$\tau_i \frac{\mathrm{d}v_i}{\mathrm{d}t} = -v_i + \sum_{j=1}^{n} W_{ij} \ g(v_j) + h_i \ , \tag{3}$$

where $\tau_i$ is the membrane time constant, and $h_i$ is a constant external input, independent of the memory we wish to recall.

It is worth noting that, since the gain function $g(v_i)$ defined in Eq (1) has no upper saturation, recurrent interactions can easily result in runaway excitation and firing rates growing unbounded. However, our optimization algorithm will naturally seek stable solutions, in which firing rates are kept within a limited range due to a fine dynamic balance of excitation and inhibition [14].

**Optimizing network parameters to embed attractor memories**

We are going to build and study networks that have a desired set of analog activity patterns as *stable fixed points* of their dynamics. Let $\{\mathbf{v}_{\text{exc}}^{\mu}\}_{\mu=1,\ldots,m}$ be a set of $m$ target analog patterns (Fig. 1), defined in the space of excitatory neuronal activity (potentials). For a given pattern $\mu$, the inhibitory neurons will be free to adjust their steady state firing rates $\mathbf{v}_{\text{inh}}^{\mu}$ to whatever pattern proves to be optimal to maintain stability. In other words, we think of the activity of inhibitory neurons as "auxiliary" variables.

A given activity pattern $\mathbf{v}^{\mu} \equiv (\mathbf{v}_{\text{exc}}^{\mu\top}, \mathbf{v}_{\text{inh}}^{\mu\top})^{\top}$ is a *stable fixed point* of the network dynamics if, and only if, it satisfies the following two conditions:

$$\left. \frac{\mathrm{d}\mathbf{v}}{\mathrm{d}t} \right|_{\mathbf{v}=\mathbf{v}^{\mu}} = 0 \qquad \text{and} \qquad \alpha\left(\mathbf{J}^{\mu}\right) < 0 \tag{4}$$

where $\mathbf{J}^{\mu}$ is the Jacobian matrix of the dynamics in Eq. 3, i.e. $J_{ij}^{\mu} := W_{ij} \ g'(v_j^{\mu}) - \delta_{ij}$ (Kronecker's delta), and $\alpha(\mathbf{J}^{\mu})$ denotes the spectral abscissa (SA), defined as the largest real part in the eigenvalue spectrum of $\mathbf{J}^{\mu}$. The first condition makes $\mathbf{v}^{\mu}$ a fixed point of the dynamics, while the second condition makes that fixed point asymptotically stable with respect to small local perturbations. Note that the width of the basin of attraction is not captured by the SA.

The two conditions in Eq. 4 depend on a set of network parameters that we will allow ourselves to optimize. These are all the synaptic weight parameters ($\beta_{ij}, i \neq j$), as well as the values of the inhibitory neurons' firing rates in each attractor ($\mathbf{v}_{\text{inh}}^{\mu}, \mu = 1, \ldots, m$). Thus, we may adjust a total of $n(n-1) + n_{\text{I}} m$ parameters.

Using Eq. 3, the first condition in Eq. 4 can be rewritten as $v_i^{\mu} - \sum_{j=1}^{n} W_{ij} g(v_j^{\mu}) - h_i = 0$. Despite this equation being linear in the synaptic weights, the re-parameterization of Eq. 2 makes it nonlinear in $\boldsymbol{\beta}$, and it is in any case nonlinear in $\mathbf{v}_{\text{inh}}^{\mu}$. We will therefore seek to satisfy this condition by minimizing $\| \mathrm{d}\mathbf{v}/\mathrm{d}t|_{\mathbf{v}=\mathbf{v}^{\mu}} \|^2$, which quantifies how fast the potentials drift away when initialized in the desired attractor state $\mathbf{v}^{\mu}$. When it is zero, $\mathbf{v}^{\mu}$ is a fixed point of the dynamics. Our optimization procedure (see below) may not be able to set this term to exactly zero, especially as we try to store a large number of memories, but in practice we find it becomes small enough that the Jacobian-based stability criterion remains valid.

Meeting the stability condition (second condition in Eq. 4) turns out to be more involved. The SA is, in general, a non-smooth function of the matrix elements and is therefore difficult to minimize.

A more suitable stability measure has been introduced recently in the context of robust control theory [13, 14], called the Smoothed Spectral Abscissa (SSA), which we will use here and denote by $\tilde{\alpha}_\varepsilon(\mathbf{J}^\mu)$. The SSA, defined for some smoothness parameter $\varepsilon > 0$, is a differentiable relaxation of the SA, with the properties $\alpha(\mathbf{J}^\mu) < \tilde{\alpha}_\varepsilon(\mathbf{J}^\mu)$ and $\lim_{\varepsilon \to 0} \tilde{\alpha}_\varepsilon(\mathbf{J}^\mu) = \alpha(\mathbf{J}^\mu)$. Therefore, the criterion $\tilde{\alpha}_\varepsilon(\mathbf{J}^\mu) \leq 0$ implies $\alpha(\mathbf{J}^\mu) < 0$, and can therefore be used as an indication of local stability. Both the SSA and its gradient are straightforward to evaluate numerically, making it amenable to minimization through gradient descent. Note that the SSA depends on the Jacobian matrix elements $\{J_{ij}^\mu\}$, which in turn depend both on the connectivity parameters $\{\beta_{ij}\}$ *and* on $\mathbf{v}_{\text{inh}}^\mu$. Note also that the parameter $\varepsilon > 0$ controls how tightly the SSA hugs the SA. Small values make it a tight upper bound, with increasingly ill-behaved gradients. Large values imply more smoothness, but may no longer guarantee that the SSA has a negative minimum even though the SA might have one. In our system of $n = 150$ neurons we found $\varepsilon = 0.01$ to yield a good compromise. In the general case the distance between SA and SSA grows linearly with the number of dimensions. To keep it invariant, $\varepsilon$ should be scaled accordingly. We therefore used the following heuristic rule $\varepsilon = 0.01 \cdot 150/n$.

We summarize the above objective into a global cost function by lumping together the fixed point and stability conditions, summing over the entire set of $m$ target memory patterns, and adding an $L_2$ penalty term on the synaptic weights to regularize:

$$\psi\left(\{\beta_{ij}\}, \{\mathbf{v}_{\text{inh}}^\mu\}\right) \coloneqq \frac{1}{m}\sum_{\mu=1}^m \left(\frac{1}{n}\left\|\frac{d\mathbf{v}}{dt}\right\|_{\mathbf{v}=\mathbf{v}^\mu}^2 + \eta_s\tilde{\alpha}_\varepsilon\left(\mathbf{J}^\mu\right)\right) + \frac{\eta_{\text{F}}}{n^2}\left\|\mathbf{W}\right\|_{\text{F}}^2 \quad . \tag{5}$$

where $\|\mathbf{W}\|_{\text{F}}^2$ is the squared Frobenius norm of $\mathbf{W}$, i.e. the sum of its squared elements, and the parameters $\eta_s$ and $\eta_{\text{F}}$ control the relative importance of each component of the objective function. We set them heuristically (Table 1). We used a variant of the low-storage BFGS algorithm included in the open source library NLopt [17] to minimize $\psi$.

**Choice of initial parameters and attractors**

The synaptic weights are initially drawn randomly from a Gamma distribution with a shape factor of 2 and a mean that depends only on the type of pre- and post-synaptic population. The mean synaptic weights of the four synapse types were computed using a mean-field reduction of the full network to meet the condition that the network initially exhibits a stable baseline state $\mathbf{v}_{\text{exc}}^{\mu=1}$ in which all excitatory firing rates equal $r_{\text{baseline}} = 5$ Hz (Table 1, and Supplementary Material). This baseline state was included in every set of $m$ target attractors that we used and was thus stable from the beginning, by construction. For the remaining target patterns, $\{\mathbf{v}_{\text{exc}}^\mu\}_{\mu=2,\dots,m}$ were generated by inverting (using $g^{-1}$) firing rates that were sampled from a log-normal distribution with a mean matching the baseline firing rate, $r_{\text{baseline}}$ (Fig. 1a) and a variance of 5 Hz. This log-normal distribution was chosen to roughly capture the skewed and heavy-tailed nature of firing rate distributions observed *in vivo* (see e.g. for a review [18]). The inhibitory potentials in the memory states, $\{\mathbf{v}_{\text{inh}}^\mu\}$, were initialized to the baseline, $g^{-1}(5\,\text{Hz})$, and were subsequently used as free parameters by the learning algorithm (cf. above; see also Fig. 1b).

## 3  Results

**Example of successful storage**

Figure 2 shows an example of stability optimization: in this specific run we used 150 neurons to embed 30 graded attractors (examples of which where shown in Fig. 1), yielding a storage capacity of 0.2. Other parameters are listed in Table 1. Gradient descent gradually reduces each of the attractor-specific sub-objectives in Eq. 5, namely the SSA, the SA, and the potential velocities $\|d\mathbf{v}/dt\|^2$ in each target state (Fig. 2). After convergence, the SSA has become negative for all desired states, indicating stability. Note, however, that $\|d\mathbf{v}/dt\|$ after convergence is small but non-zero in each of the target memories. Thus, strictly speaking, the target patterns haven't become fixed points of the dynamics, but only slow points from which the system will eventually drift away. In practice though, we found that stability was robust enough that an exact, stable fixed point had in fact been created very near each target pattern. This is detailed below.

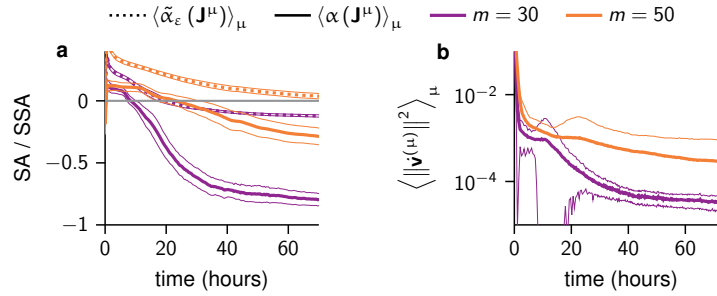

Figure 2: **(a)** Decrease of the SA (solid line) and of the SSA (dotted line) during learning in systems with 30 (purple) and 50 attractors (orange). Thick lines show averages across attractors, flanking lines show the corresponding standard deviations. The x-axis marks the actual duration of the run of the learning algorithm. **(b)** Euclidean norm of the velocity at the fixed point during learning. Lines and colors as in **a**. Note the logarithmic y-axis.

Table 1: Parameter settings

| $n_{\mathrm{E}}$ | 100 | $\tau_{\mathrm{E}}$ | 20 ms | $\eta_{\mathrm{s}}$ | 0.02 |
|---|---|---|---|---|---|
| $n_{\mathrm{I}}$ | 50 | $\tau_{\mathrm{I}}$ | 10 ms | $\eta_{\mathrm{F}}$ | 0.001 |
| $m$ | 30 | $r_{\mathrm{baseline}}$ | 5 Hz | | |

**Memory recall performance and robustness**

For recall, we initialize neuronal activities at a noisy version of one of the target patterns, and study the subsequent evolution of the network state. The network performs well if its dynamics clean up the noise and home in on the target pattern (autoassociative behavior) and if it achieves this robustly even in the face of large amounts of noise.

Initial cues are chosen to be linear combinations of the form $\mathbf{r}(t=0) = \sigma\,\tilde{\mathbf{r}} + (1-\sigma)\,\mathbf{r}^\mu$, where $\mathbf{r}^\mu$ is the memory we intend to recall and $\tilde{\mathbf{r}}$ is an independent random vector with the same lognormal statistics used to generate the memory patterns themselves. The parameter $\sigma$ regulates the noise level: $\sigma = 0$ sets the network activity directly in the desired attractor, while $\sigma = 1$ initializes it with completely random values.

The deviation of the momentary network state $\mathbf{r}(t) \equiv g(\mathbf{v}(t))$ from the target pattern $\mathbf{r}^\mu \equiv g(\mathbf{v}^\mu)$ is measured in terms of the squared Euclidean distance, further normalized by the expected squared distance between $\mathbf{r}^\mu$ and a random pattern drawn from the same distribution (log-normal in our case). Formally:

$$d_\mu(t) := \frac{\|\mathbf{r}_{\mathrm{exc}}(t) - \mathbf{r}_{\mathrm{exc}}^\mu\|^2}{\langle\|\tilde{\mathbf{r}}_{\mathrm{exc}} - \mathbf{r}_{\mathrm{exc}}^\mu\|^2\rangle_{\tilde{\mathbf{r}}}}. \tag{6}$$

Figure 3a shows the temporal evolution of $d_\mu(t)$ on a few sample recall trials, for two different noise levels $\sigma$. For $\sigma = 0.5$, recalls are always successful, as the network state converges to the right target pattern on each trial. For $\sigma = 0.75$, the network activity occasionally settles in another, well distinct attractor.

We used the convention that a trial is deemed successful if the distance $d_\mu(t)$ falls below 0.001. (A $\sim 3$ Hz deviation from the target in only one of the 100 exc. neurons, with all other 99 neurons behaving perfectly, would be sufficient to cross this threshold and fail the test.) We further measure performance as the probability of successful recall, which we estimated from many independent trials with different realizations of the noise $\tilde{\mathbf{r}}$ in the initial condition (Figure 3b). The network performance is also compared to an "ideal observer" [6] that has direct access to all the stored memories (rather than just their reflection in the synaptic weights) and simply returns that pattern in the training set $\{\mathbf{r}^\mu\}$ to which the initial cue is closest (Fig. 3b). Thus, as an upper bound on performance, the ideal observer only produces a wrong recall when the added noise brings the initial state closer to an attractor that is different from the target. Remarkably, our network dynamics

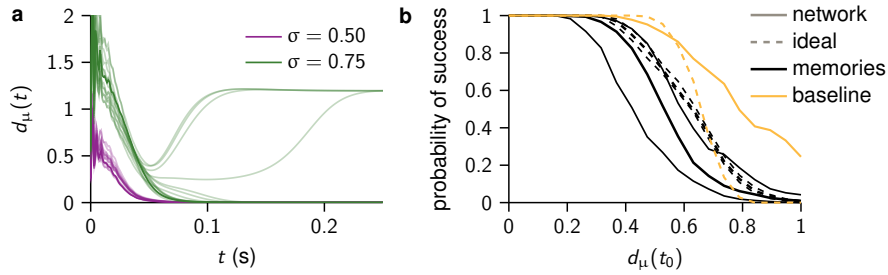

Figure 3: **(a)** Example recall trials for a single memory $\mathbf{r}^\mu$, which is presented to the network at time $t = 0$ in a corrupted version that is different on every trial, for two different values of the noise level $\sigma$ (colors). Shown here is the temporal evolution of the momentary distance between the vector of excitatory firing rates $\mathbf{r}_{\mathrm{exc}}(t)$ and the memory pattern $\mathbf{r}_{\mathrm{exc}}^\mu$. Different lines correspond to different trials. **(b)** Fraction of trials that converged onto the correct attractor (final distance $d_\mu(t = \infty) < 0.001$, cf. text) as a function of the normalized distance between the initial condition and the desired attractor, $d_\mu(t = 0)$. Thick lines show medians across attractors, flanking thin lines show the 25$^{\text{th}}$ and 75$^{\text{th}}$ percentiles. The performance of the baseline state is shown separately (orange). The dashed lines show the performance of an "ideal observer", always selecting the memory closest to the initial condition, for the same trials.

(continuous lines) and the ideal observer (dashed lines) have comparable performances. When trying to recall the uniform pattern of baseline activity, the performance appears much better (orange line) both for the ideal observer and the network. This is simply because the random vectors used to perturb the system have a high probability of lying closer to the mean of the log normal distribution (that is, the baseline state) than to any other memory pattern. Moreover, the network was initialized prior to learning with the baseline as the single global attractor, and this might account for the additional tendency of the network (solid orange line) to fall on such state, as compared to the ideal observer (dotted orange line).

**Only a few strong synaptic weights contribute to memory recall**

Synaptic weights after learning (Fig. 4a) are sparse: their distribution shows the characteristic peak near zero and the long tail observed in real cortical circuits [19, 20] (Fig. 4b). This sparseness cannot be accounted for by the $L_2$ norm regularizer in the cost function (Eq. 5) as it does not promote sparsity as an $L_1$ term would. Thus, the observed sparsity in the trained network must be a genuine consequence of having optimized the connectivity for robust stability.

If we assume that weights $|W_{ij}| \leq 0.01$ correspond to functionally silent synapses, then the trained network contains 52% of silent excitatory synapses and 46% of silent inhibitory ones (Fig. 4c). We wondered if those weak, "silent" synapses are necessary for stability of memory recall, or could be removed altogether without affecting performance. To test that, we clipped those synapses $\{|W_{ij}| < 0.01\}$ to zero, and computed recall performance again (Fig. 4d). This clipping turns out to slightly shift the position of the attractors in state space, so we increased the distance threshold that defines a successful recall trial to 0.08. The test reveals that one of the attractors loses stability, reducing the average performance. However the remaining 29 attractors are robust to this removal of weak synapses and show near-equal recall performance as above. This demonstrates that small weights, though numerous, are not necessary for competent recall performance.

**Balanced state**

As a result of the connection weight distributions and robust stability, the trained network produces a regime in which excitation and inhibition balance each other, precisely tuning each neuron to its target frequency in each attractor. Excitatory and inhibitory inputs are defined as $h_i^{\mathrm{exc}}(t) = \sum_{j=1}^n \lfloor W_{ij} \rfloor_+ \, r_j(t)$ and $h_i^{\mathrm{inh}}(t) = \sum_{j=1}^n \lfloor -W_{ij} \rfloor_+ \, r_j(t)$ so that the difference $h_i^{\mathrm{exc}}(t) - h_i^{\mathrm{inh}}(t)$ corresponds to the total recurrent input, i.e. the second term on the r.h.s. of Eq. 3.

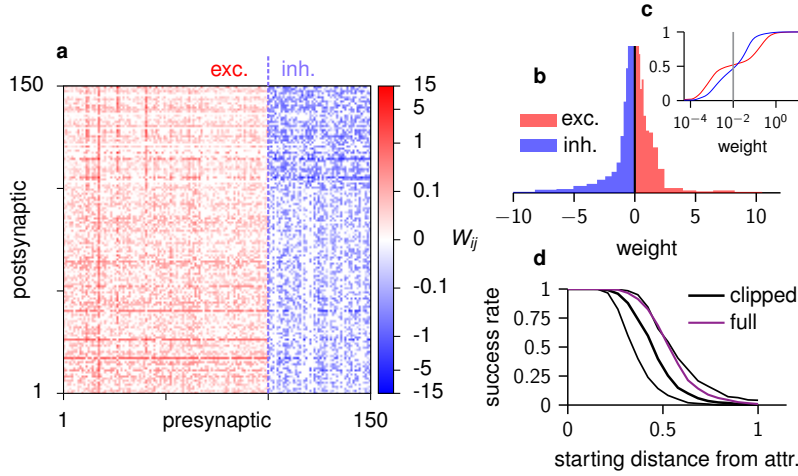

Figure 4: **(a)** Synaptic weight matrix after learning. Note the logarithmic color scale. **(b)** Distribution of the excitatory (red) and inhibitory (blue) weights. **(c)** Cumulative weight distribution of absolute weight values. Gray line marks the 0.01 threshold we use to defined "silent" synapses. **(d)** Performance of the network after clipping the weights below 0.01 to zero (black, median with 25[th] and 75[th] percentiles), compared to the performance of the unperturbed network redrawn from Fig. 3 (purple).

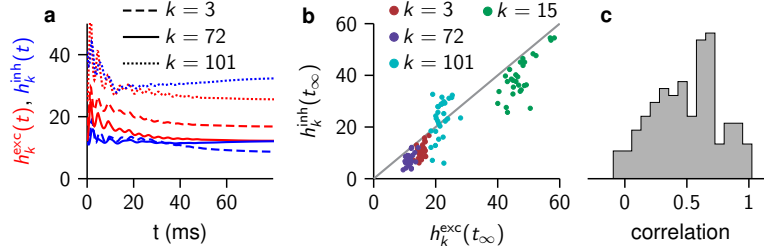

Figure 5: **(a)** Dynamics of the excitatory and inhibitory inputs during a memory recall trial, for three sample neurons. **(b)** Scatter plot of steady-state excitatory versus inhibitory inputs. Each dot corresponds to a different memory pattern, and several neurons are shown in different colors. **(c)** Histogram of E and I input correlations across all memories for each neuron (for example, one value binned in this histogram would be the correlation between all green dots in **b**).

Figure 5a shows the evolution of $h_i^{\text{exc}}(t)$ and $h_i^{\text{inh}}(t)$ during a recall trial for one of the stored random attractors, for 3 different neurons. Neuron 3 has rate target of 9Hz, well above average, therefore its excitation is much higher than inhibition. Neuron 72 has a steady state firing rate of 2 Hz, below average: its inhibitory input is greater than the excitatory one, and firing is driven by the external current. Finally, neuron 101 is inhibitory and has a target rate 0, and indeed its inhibitory input is large enough to overwhelm the combined effects of the external and recurrent excitatory inputs. Notably, in all these cases, both E and I input currents are fairly large but cancel each other to leave something smaller, either positive or negative.

Figure 5b shows the E vs. I inputs at steady-state across all the embedded attractors, for various neurons plotted in different colors. These E and I inputs tend to be correlated across attractors for every single neuron (dots in Fig. 5 tend to hug the identity line), with relative differences fine-tuned to yield the desired firing rates. These across-attractors E/I correlations are summarized in Fig. 5c as a histogram over neurons.

## Robustness to ongoing noise and reduction of across-trial variability following recall onset

Finally, to probe the system under more realistic dynamics, we added time-varying, Gaussian white noise such that, in an excitatory neuron free from network interactions, the potential would fluctuate

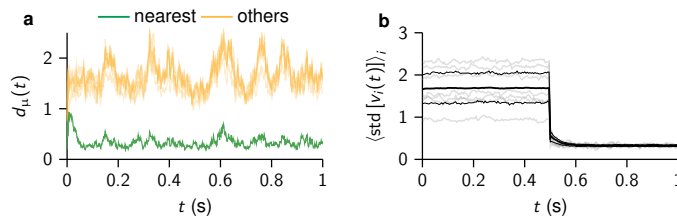

Figure 6: **(a)** Normalized distance calculated according to Eq. 6 between the network activity and each of the attractors (targeted attractor: green line; others: orange lines) during a noisy recall episode. **(b)** Trial-to-trial variability, expressed as the standard deviation of a neuron's activity across multiple repetitions with random initial conditions. At time $t = 0.5$ s the network receives a pulse in the direction of one target attractor ($\mu = 2$). Gray lines are for single neurons; the black line is an average over the population.

with standard deviation 0.33. Figure 6a shows the momentary distance $d_\mu(t)$ of the network state from the attractor closest to the initial cue (green), and for all other attractors (orange), during a recall trial. It is clear that the system revolves around the desired attractor, performing successful recall despite the ongoing noise. In a second experiment, we ran many trials in which the initialization at time $t = 0$ was random, while the same spatially patterned stimulation – aligned onto a chosen attractor – is given to the network in each trial at time $t = 0.5$ sec. Figure 6b shows the standard deviation of the internal state of a neuron across trials, averaged across the neural population. Following stimulus onset, neurons are always pushed towards the target attractor, and this greatly reduces trial-by-trial variability, compared to the initial spontaneous regime in which the neurons would fluctuate around any of the activity levels corresponding to its assigned attractors. Interestingly, such stimulus-induced variability reduction has been observed very broadly across sensory and motor cortical areas [21]. This extends previous work, e.g. [22] and [23], showing variability reduction in a multiple-attractor scenario with effectively binary patterns, to the case of patterns with graded activities.

## 4    Discussion

We have provided a proof of concept that a model cortical networks of E and I neurons can embed multiple analog memories as stable fixed-points of their dynamics. Memories are stable in the face of ongoing noise and corruption of the recall cues. Neuronal activities do not saturate, and indeed, our single-neuron model did not explicitly incorporate an upper saturation mechanism: dynamic feedback inhibition, precisely matched to the level of excitation incurred by each attractor, ensures that each neuron can fire at a relatively low rate during recall. As a result, excitation and inhibition are tightly balanced.

We have used a rate-based formulation of the circuit dynamics, which raises the question of the applicability of our method to understanding spiking memory networks. Once the connectivity in the rate model is generated and optimized, it could still be used in a spiking model, provided the gain function we have used here matches that of the single spiking neurons. In this respect, the gain function we have used here is likely an appropriate choice: in physiological conditions, cortical neurons have input-output gain functions that are well approximated by a rectified power-law function over their entire dynamic range [24, 25, 26].

An important question for future research is how local synaptic learning rules can achieve the stabilization objective that we have approached here from an optimal, algorithmic viewpoint. Inhibitory synaptic plasticity is a promising candidate, as it has already been shown to enable self-regulation of the spontaneous, baseline activity regime, and also to promote the stable storage of binary memory patterns [27]. More work is required in this direction.

**Acknowledgements.** This work was supported by the Wellcome Trust (GH, ML), the European Union Seventh Framework Programme (FP7/20072013) under grant agreement no. 269921 (BrainScaleS) (DF, ML), and the Swiss National Science Foundation (GH).

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
