[Supplementary Material]

# Analog Memories in a Balanced Rate-Based Network of E-I Neurons
# — Supplementary Material —

**Dylan Festa**
df325@cam.ac.uk

**Guillaume Hennequin**
gjeh2@cam.ac.uk

**Máté Lengyel**
m.lengyel@eng.cam.ac.uk

Computational & Biological Learning Lab, Department of Engineering
University of Cambridge, UK

## 1 Weight initialization

The initial weights prior to learning are set randomly, but with the requirement that the baseline rate is already a stable attractor of the system. To achieve this, we first consider the system in terms of average population activities, i.e. reducing it to 2 dimensions, one representing excitatory (E) neurons and the other representing inhibitory (I) ones.

$$\begin{pmatrix} \tau_{\mathrm{E}} & 0 \\ 0 & \tau_{\mathrm{I}} \end{pmatrix} \frac{\mathrm{d}}{\mathrm{d}t} \begin{pmatrix} v_{\mathrm{E}} \\ v_{\mathrm{I}} \end{pmatrix} = - \begin{pmatrix} v_{\mathrm{E}} \\ v_{\mathrm{I}} \end{pmatrix} + \begin{pmatrix} W_{\mathrm{EE}}^{\mathrm{2D}} & -W_{\mathrm{EI}}^{\mathrm{2D}} \\ W_{\mathrm{IE}}^{\mathrm{2D}} & -W_{\mathrm{II}}^{\mathrm{2D}} \end{pmatrix} \begin{pmatrix} g(v_{\mathrm{E}}) \\ g(v_{\mathrm{I}}) \end{pmatrix} + \begin{pmatrix} h \\ h \end{pmatrix} \tag{S1}$$

We fix the time constants to $\tau_{\mathrm{E}} = 20$ ms and $\tau_{\mathrm{I}} = 10$ ms, leaving a total of five free parameters (all positive). The stationary points of the network solve the equation:

$$\begin{pmatrix} \tilde{v}_{\mathrm{E}} \\ \tilde{v}_{\mathrm{I}} \end{pmatrix} = \begin{pmatrix} W_{\mathrm{EE}}^{\mathrm{2D}} & -W_{\mathrm{EI}}^{\mathrm{2D}} \\ W_{\mathrm{IE}}^{\mathrm{2D}} & -W_{\mathrm{II}}^{\mathrm{2D}} \end{pmatrix} \begin{pmatrix} g(\tilde{v}_{\mathrm{E}}) \\ g(\tilde{v}_{\mathrm{I}}) \end{pmatrix} + \begin{pmatrix} h \\ h \end{pmatrix} \tag{S2}$$

In addition, we have to ensure that this activity level is stable. The necessary and sufficient condition for this is that the Jacobian calculated in the fixed point has negative eigenvalues. The Jacobian matrix of the reduced model is:

$$\mathbf{J}|_{\mathbf{v}=\tilde{\mathbf{v}}} = \begin{pmatrix} \tau_{\mathrm{E}} & 0 \\ 0 & \tau_{\mathrm{I}} \end{pmatrix}^{-1} \left[ -\mathbf{I} + \mathbf{W}^{\mathrm{2D}} \begin{pmatrix} g'(\tilde{v}_{\mathrm{E}}) & 0 \\ 0 & g'(\tilde{v}_{\mathrm{I}}) \end{pmatrix} \right] \tag{S3}$$

The conditions for having only negative eigenvalues in a 2-dimensional $\mathbf{J}$ are:

$$\mathrm{Tr}(\mathbf{J}|_{\mathbf{v}=\tilde{\mathbf{v}}}) < 0 \qquad \text{and} \qquad \mathrm{Det}(\mathbf{J}|_{\mathbf{v}=\tilde{\mathbf{v}}}) > 0 \ . \tag{S4}$$

We picked values for our 6 free parameters in order to satisfy Eq. S2 and Eq. S4 within some margin for reasonably low values of $\tilde{v}_{\mathrm{E}}$ and $\tilde{v}_{\mathrm{I}}$ (many combinations of values would work and promote successful learning in the full-size system). The specific parameters used for the simulations are:

$$\mathbf{W}^{\mathrm{2D}} = \begin{pmatrix} 2.5 & -1.3 \\ 2.4 & -1 \end{pmatrix} \qquad h = 7 \qquad g(\tilde{\mathbf{v}}) = \begin{pmatrix} r_{\mathrm{E},baseline} \\ r_{\mathrm{I},baseline} \end{pmatrix} \approx \begin{pmatrix} 5 \\ 6.5 \end{pmatrix} \tag{S5}$$

To build the full-scale N-dimensional weight matrix, we sampled random values from an i.i.d. gamma distribution with a shape parameter of 2 and a mean that depended on the neural populations involved. If neuron $i$ belongs to population $\alpha$ and neuron $j$ belongs to population $\beta$ (where $\alpha, \beta \in \{\mathrm{E}, \mathrm{I}\}$) the distribution mean is $\langle W_{ij} \rangle = W_{\alpha\beta}/n_{\beta}$. To avoid fluctuations in the mean incoming weight on a single-cell basis, we further enforced the following normalization: $W_{ij} \leftarrow W_{ij} \frac{W_{\alpha\beta}^{\mathrm{2D}}}{\sum_{j \in \{\beta\}} W_{ij}}, \quad \forall i \in \{\alpha\}$. Finally, the constant inputs $h_i$ in the large system were all set to $h$.

Building the network in this way ensures that a baseline state of uniform, low firing rates across the network is initially stable.

## 2 Recap: system dynamics and the Jacobian

The dynamics of the system is expressed by Eq. 3 of the main text, repeated here for convenience:

$$\frac{\mathrm{d}\,v_i}{\mathrm{d}t} = \frac{1}{\tau_i}\left(-v_i + \sum_{j=1}^{n} W_{ij}\,g(v_j) + h_i\right) \tag{3}$$

To constrain the signs of the synaptic weight and thus enforce Dale's law, we reparameterized the weights as

$$W_{ij} = (1 - \delta_{ij})\,s_j\,\log(1 + \exp\beta_{ij}) \quad \text{with} \quad s_j = \begin{cases} +1 & \text{if} \quad j \le n_{\mathrm{E}} \\ -1 & \text{if} \quad \text{otherwise} \end{cases} \tag{2}$$

where the $(1 - \delta_{ij})$ term prevents the existence of autapses. The single-neuron I/O gain function is threshold-quadratic:

$$g(v_i) = \gamma\,\lfloor v_i \rfloor_+^2\,, \quad \text{with} \quad \lfloor x \rfloor_+ = \begin{cases} x & \text{if} \quad x > 0 \\ 0 & \text{otherwise} \end{cases} \tag{1}$$

The $(i, j)^{\text{th}}$ element of the Jacobian matrix $\mathbf{J}$ is:

$$J_{ij} := \frac{\partial}{\partial v_j}\left\{\frac{\mathrm{d}\,\mathbf{v}}{\mathrm{d}t}\right\}_i = \frac{1}{\tau_i}\left(-\delta_{ij} + W_{ij}\,g'(v_j)\right) \tag{S6}$$

## 3 Computation of the smoothed spectral abscissa (SSA) and its gradient

In this section we summarize the procedure employed to compute the SSA of the Jacobian, and its gradient with respect to each matrix element. For a more thorough description, see the original paper introducing the SSA [1]; for a more specific application to neuroscience and interpretation in terms of network dynamics, see [2].

Given a square matrix $\mathbf{J}$, our goal is to compute its SSA, denoted by $\tilde{\alpha}_\varepsilon(\mathbf{J})$. For this purpose, a function $f : (\mathbb{R}^{n \times n}, \mathbb{R}) \to \mathbb{R}$ is defined as follows:

$$f(\mathbf{J}, s) = \mathrm{Tr}(\mathbf{P}_s) \tag{S7}$$

where $\mathbf{P}_s$ satisfies the following Lyapunov equation:

$$(\mathbf{J} - s\mathbf{I})\mathbf{P}_s + \mathbf{P}_s(\mathbf{J} - s\mathbf{I})^{\mathrm{T}} = -\mathbf{I} \tag{S8}$$

In Matlab, such equations can be solved using the `lyap` function, which uses the standard Bartels-Stewart algorithm [3]. It is also convenient to define a second matrix $\mathbf{Q}_s$ as the solution to the Lyapunov equation dual to the previous one:

$$(\mathbf{J} - s\mathbf{I})^{\mathrm{T}}\mathbf{Q}_s + \mathbf{Q}_s(\mathbf{J} - s\mathbf{I}) = -\mathbf{I} \tag{S9}$$

The SSA, also denoted by $\tilde{\alpha}_\varepsilon(\mathbf{J})$, corresponds to the scalar $s$ that solves $f(\mathbf{J}, s) = 1/\varepsilon$ for some $\varepsilon > 0$. There are no closed-form solutions, but the smoothness of $f(\mathbf{J}, s)$ makes it possible to use standard root-finding methods, requiring the gradient w.r.t. $s$:

$$\frac{\partial f(\mathbf{J}, s)}{\partial s} = -2\,\mathrm{Tr}(\mathbf{Q}_s\mathbf{P}_s) \tag{S10}$$

Here, we used the standard Newton method to find the root of $g(s) = f(\mathbf{J}, s) - 1/\varepsilon$, yielding the SSA. Finally, once the value of $\tilde{\alpha}_\varepsilon(\mathbf{J})$ is known, its gradient w.r.t. $\mathbf{J}$ can be computed as follows:

$$\frac{\partial \tilde{\alpha}_\varepsilon(\mathbf{J})}{\partial \mathbf{J}} = \frac{\mathbf{Q}_{\tilde{\alpha}_\varepsilon}\mathbf{P}_{\tilde{\alpha}_\varepsilon}}{\mathrm{Tr}(\mathbf{Q}_{\tilde{\alpha}_\varepsilon}\mathbf{P}_{\tilde{\alpha}_\varepsilon})} \tag{S11}$$

**Complexity** The main computational bottleneck in computing the SSA and its gradient is solving the Lypunov equations, which is $\mathcal{O}(n^3)$. It is worth noting that substantial acceleration can be achieved by computing the Schur decomposition of $\mathbf{J}$ *before* starting iterating through the Newton algorithm. Indeed, the Schur decomposition is the most expensive ($\mathcal{O}(n^3)$) part of the Bartels-Stewart algorithm, and once it is computed, solving the Lyapunov equation takes only $\mathcal{O}(n^2)$ operations. Moreover, the upper-triangular Schur factor is common to all shifted versions $\mathbf{J} - s\mathbf{I}$, so it only needs to be computed once.

# 4 The gradient of the cost function

The cost function is defined in Eq. 5 in the main text; we repeat it here for conveninence:

$$\psi\left(\{\beta_{ij}\}, \{\mathbf{v}^\mu\}\right) = \sum_{\mu=1}^{m} \left(\frac{1}{n} \, \|\dot{\mathbf{v}}\|^2_{\mathbf{v}=\mathbf{v}^\mu} + \eta_\mathrm{s} \, \tilde{\alpha}_\varepsilon\left(\mathbf{J}^\mu\right)\right) + \frac{\eta_\mathrm{F}}{n^2} \, \|\mathbf{W}\|^2_\mathrm{F} \tag{5}$$

where $\dot{\mathbf{v}} \equiv \mathrm{d}\mathbf{v}/\mathrm{d}t$.

We show here how to compute the gradient of Eq. 5 with respect to the parameters we are optimising over. These parameters consist of the weight parameters $\beta_{ij}$ and the activity $v_i^\mu$ of each auxiliary neuron $i > n_\mathrm{E}$ in each attractor $\mu$. In our simulations, we used $n_\mathrm{E} = 100$, $n_\mathrm{I} = 50$ and 30 memories, yielding a total number of free parameters of $150^2 + 30 \cdot 50 = 24000$.

We now proceed to compute the derivatives of each of the terms in $\psi$.

## 4.1 Fist term: velocity $\|\dot{\mathbf{v}}\|^2$

To simplify the notations, we consider a single attractor, and thus drop the $\cdot^\mu$ superscript. We start with the derivatives with respect to the weight parameters. Note that $W_{ij}$ depends only on $\beta_{ij}$, so the application of the chain rule is straightforward:

$$\frac{\partial \|\dot{\mathbf{v}}\|^2}{\partial \beta_{\ell p}} = \frac{\partial \|\dot{\mathbf{v}}\|^2}{\partial W_{\ell p}} \frac{\partial W_{\ell p}}{\partial \beta_{\ell p}} = 2 \frac{\partial W_{\ell p}}{\partial \beta_{\ell p}} \sum_i \dot{v}_i \frac{\partial \dot{v}_i}{\partial W_{\ell p}} \tag{S12}$$

The first partial derivative is solved starting from Eq. 2:

$$\frac{\partial W_{\ell p}}{\partial \beta_{\ell p}} = s_p \frac{1}{1 + \exp(-\beta_{\ell p})} = s_p(1 - \exp(-|W_{\ell p}|)) \tag{S13}$$

The second partial derivative is solved using Eq. 3:

$$\sum_i \dot{v}_i \frac{\partial \dot{v}_i}{\partial W_{\ell p}} = \sum_i \frac{\dot{v}_i}{\tau_i} \left(\sum_j \delta_{i\ell} \, \delta_{jp} \, g(v_j)\right)$$

$$= \frac{\dot{v}_\ell}{\tau_\ell} \, g(v_p) \tag{S14}$$

Due to the absence of autapses, the value of Eq. S14 should be null for $\ell = p$. We can safely ignore this condition as the value of Eq. S13 is zero for null weights, so that the product in Eq. S12 is also zero.

The derivative with respect to the activity of each auxiliary neuron $\ell$ (again dropping the $\cdot^\mu$ superscript for convenience) is the following:

$$\frac{\partial \|\dot{\mathbf{v}}\|^2}{\partial v_\ell} = 2 \sum_i \dot{v}_i \frac{\partial \dot{v}_i}{\partial v_\ell}$$

$$= 2 \sum_i \frac{\dot{v}_i}{\tau_i} \left(-\delta_{i\ell} + \sum_j W_{ij} \, \delta_{j\ell} \, g'(v_\ell)\right)$$

$$= 2 \sum_i \frac{\dot{v}_i}{\tau_i} \left(-\delta_{i\ell} + W_{i\ell} \, g'(v_\ell)\right) \tag{S15}$$

## 4.2 Second term: SSA of the Jacobian

The Jacobian matrix is given by Eq. S6. Section 3 illustrates how the SSA and all the partial derivatives are computed. Our free parameters, however, are not the $J_{ij}$ terms: calculating the gradient

with respect to weight and rate parameters requires the use of the chain rule. Once again, for simplicity, we consider a single attractor, and we start from the derivative with respect to the weight parameters:

$$\frac{\partial \tilde{\alpha}_\varepsilon (\mathbf{J})}{\partial \beta_{\ell p}} = \sum_{ij} \frac{\partial \tilde{\alpha}_\varepsilon (\mathbf{J})}{\partial J_{ij}} \frac{\partial J_{ij}}{\partial W_{\ell p}} \frac{\partial W_{\ell p}}{\partial \beta_{\ell p}} \tag{S16}$$

The fist term is given by Eq. S11, the third term has already been computed in Eq. S13, so only the middle term needs to be computed. Using Eq. S6:

$$\frac{\partial J_{ij}}{\partial W_{\ell p}} = \frac{1}{\tau_i} \delta_{i\ell} \delta_{jp} g'(v_j) \tag{S17}$$

For the no-autapses condition, this value should be zero for $l = m$, but once again this condition is covered by the third term of Eq. S16. By virtue of the delta-functions, the sum in Eq S16 simplifies as follows:

$$\frac{\partial \tilde{\alpha}_\varepsilon (\mathbf{J})}{\partial \beta_{\ell p}} = \frac{1}{\tau_\ell} g'(v_p) \frac{\partial \tilde{\alpha}_\varepsilon (\mathbf{J})}{\partial J_{\ell p}} \frac{\partial W_{\ell p}}{\partial \beta_{\ell p}} \tag{S18}$$

The derivative w.r.t. the activity of each auxiliary neuron $\ell > n_\mathrm{E}$ is as follows:

$$\frac{\partial \tilde{\alpha}_\varepsilon (\mathbf{J})}{\partial v_\ell} = \sum_{ij} \frac{\partial \tilde{\alpha}_\varepsilon (\mathbf{J})}{\partial J_{ij}} \frac{\partial J_{ij}}{\partial v_\ell} ; \qquad \text{using Eq. S6} \qquad \frac{\partial J_{ij}}{\partial v_\ell} = \frac{1}{\tau_i} W_{ij} \delta_{j\ell} g''(v_\ell) \tag{S19}$$

This leads to:

$$\frac{\partial \tilde{\alpha}_\varepsilon (\mathbf{J})}{\partial u_\ell} = \sum_i \frac{1}{\tau_i} W_{i\ell} g''(v_\ell) \frac{\partial \tilde{\alpha}_\varepsilon (\mathbf{J})}{\partial J_{i\ell}} \tag{S20}$$

### 4.3 Third term: weight penalty

The only term left in Eq. 5 is the penalty on large weights, expressed as the squared Frobenius norm of the $\mathbf{W}$ matrix.

$$\|W\|_\mathrm{F}^2 := \sum_{ij} W_{ij}^2 \tag{S21}$$

The only non-zero derivatives are, of course, those with respect to the weight parameters:

$$\frac{\partial \|W\|_\mathrm{F}^2}{\partial \beta_{\ell p}} = 2 W_{\ell p} \frac{\partial W_{\ell p}}{\partial \beta_{\ell p}} \tag{S22}$$

### 4.4 Complete gradients

We can now write the partial derivatives of the total cost function, Eq. 5. The derivative with respect to the weight parameters is given by Eq. S12 to S14, Eq. S18 and Eq. S22

$$\frac{\partial}{\partial \beta_{\ell p}} \psi \left( \{\beta_{ij}\}, \{\mathbf{v}^\mu\} \right) =$$

$$= 2 s_p (1 - \exp(-|W_{\ell p}|)) \left[ \frac{1}{m \tau_\ell} \sum_{\mu=1}^m \left( \frac{\dot{v}_\ell^\mu}{n} g(v_p^\mu) + \eta_s \frac{\partial \tilde{\alpha}_\varepsilon (\mathbf{J})}{\partial J_{\ell p}} g'(v_p^\mu) \right) + \frac{\eta_\mathrm{F} W_{\ell p}}{n^2} \right] \tag{S23}$$

The derivative with respect to the rate of an auxiliary neuron for a specific attractor is given by Eqs. S15 and S20:

$$\frac{\partial}{\partial v_\ell^\mu} \psi \left( \{\beta_{ij}\}, \{\mathbf{v}^\mu\} \right) =$$

$$= \frac{2}{m} \sum_i \frac{1}{\tau_i} \left( \frac{\dot{v}_i}{n} (-\delta_{i\ell} + W_{i\ell} g'(v_\ell^\mu)) + \eta_s W_{i\ell} g''(v_\ell^\mu) \frac{\partial \tilde{\alpha}_\varepsilon (\mathbf{J})}{\partial J_{i\ell}} \right) \tag{S24}$$