[Reviews · NeurIPS 2014]

Submitted by Assigned_Reviewer_7

Summary: The authors present a model of auto-associative memory in a rate-based neural network subject to a battery of biological plausible constraints. Previous models of auto-associative memory have failed to include several key features of real biological networks, namely an adherence to Dale's Law that neurons have a strictly excitatory or inhibitory effect on their projections and the observation that networks can encode memories without relying on units that simply respond at their saturation rate or respond in a binary manner. Memories are encoded in the network via synaptic modifications based on a gradient descent procedure, constrained using a recently published method for ensuring that the linearization of the dynamics around a dynamical system's fixed point is stable. The authors illustrate the effectiveness of their training procedure with simulations, noting that the trained fixed points exhibit slow network dynamics (i.e. they are close to being, but are not exactly, fixed points) and are stable, as desired. The authors note two features of their trained network that are in line with experimentally observed features of cortical networks, namely a distribution of synaptic weights centered at zero with long tails and that an average network unit received approximately equal excitatory and inhibitory synaptic input. The authors identify these features as key components to the networks success in performing robust auto-associative memory when the network is perturbed from a fixed point, even in the presence of stochastic input noise.

Review: This is a clearly-written, novel submission that will be of great interest to the community. The authors nicely review the history of training neural networks to perform auto-associative tasks, highlighting the main contributions and shortcomings of previous work. The network architecture and training procedure are both presented in a clear way and the figures are well-chosen to quickly illustrate the main results of the submission. The authors illustrate the networks performance limitations when network noise is included, which highlights the robust solution the training procedure finds.

One recommendation would be to clarify the relationship between the chosen training procedure and the stated objectives. As the author's noted other methods have been employed in the past to enforce Dale's Law or achieve graded network memories, with the latter condition seemingly more difficult to enforce. The chosen training procedure achieves the desired goal of building an auto-associative network with non-saturing units, but nothing in the text indicates that the authors chose this method because they believed it was particularly well-suited to this task. Perhaps another network training procedure would achieve the same result? Can anything be said as to why this particular method achieves that goal so effectively?

Also (although probably beyond the scope of this paper) given a rich history of training networks to perform auto-associative tasks, a comparison of how the trained network performs against previous models in terms of network capacity (number of memories as a function of network units) would surely be of interest to the community.
Summary: 1-2 sentences: Applying a recent method for ensuring stable fixed points in a dynamical system, the authors introduce a new model of auto-associative memory that conforms to several constraints of real neural networks, most notably the condition that network units are capable of encoding memories without saturating or binary responses.

Submitted by Assigned_Reviewer_8

In this manuscript, the authors design a rate-based neuronal network which is trained to hold a number of stable activity patterns, i.e. memories. The goal is to overcome several shortcomings of previous attractor models, namely violation of Dale's law and saturation of activities in the memory states. For training, the authors use a gradient descent on a cost function which minimizes activity changes in the desired attractors and the Frobenius norm of the weight matrix, and maximizes the stability of attractors. The main results are a weight distribution close to experimental findings and a balance of excitation and inhibition in the attractor states.

The writing of the manuscript is good. I found the model and the training procedure easy to understand, as well as the presentation of the results. The combination of biological constraints going into the model and the resulting features after training make this paper an interesting contribution to the understanding of attractor neural networks.
Summary: The authors train a rate-based neuronal network to hold graded and non-saturated attractors in the synaptic weights and inhibitory activities. Constraints imposed on the synapses and neurons for biological plausability lead to a balance of excitation and inhibition on the network level.

Submitted by Assigned_Reviewer_36

The paper returns to the venerable problem of associative memory in neural networks with a new fresh perspective, adding much new insight to an old problem. The authors observe that much previous work in the field ignored biological aspects such as Dale’s law, used binary memory representations in rate based models, and used non-sparse representations for memories. While all these issues have been dealt with individually in other work, the authors develop a strategy which allows them to address all issues. More concretely, they make use of a recent control-theoretic approach, based on spectral bounding techniques, to construct networks which deal with all these issues. In this approach the authors propose a cost function balancing three terms to achieve the desired goal through gradient descent. The first term in the cost contributes to the fixed point nature of the desired memories, the second term to the stability of the fixed points, and the third term encourages small weights which prevent saturation. By directly enforcing Dale’s principle in the weight parameterization, the authors are able to address all the above shortcomings of previous work, leading to a network where excitations and inhibition are naturally balanced. Interestingly, while they do not enforce saturation in the single neuron firing, this arises naturally through the solution of the optimization problem. Importantly, the authors only make use of the excitatory neurons to implement the memory (which is consistent with the fact it is usually the excitatory neurons that send output to other regions), and view the inhibitory neurons as variables to be optimized over. The authors validate their results through numerical simulations, and demonstrate its noise robustness. In addition to demonstrating the good associative memory properties of the network, the authors are also able to explain experimental findings showing that the trial to trial variability is reduced following stimulus onset (although this seems to be a general feature of associative memories with spontaneous baseline firing). This is explained through convergence to the attractor closets to the initial condition.

In summary, this is a clear and well written paper. It tackles a classic problem in neural computation using novel control-theoretic tools, and demonstrates how a biologically plausible associative memory may function. They do not propose a biologically plausible learning mechanism at this point, but hint at some recent proposals that may lead to such rules. I found the utilization of the inhibitory neurons as resources to be interesting and beneficial. Overall, the paper presents a clear, concise and important contribution to neural computation.

A few minor issues:
The authors use \gamma=0.04 in eq. (3). It would good to discuss the sensitivity of the results to this choice (although I expect it to be small)
A log-normal distribution for memory states is proposed on line 115. Please motivate this.
The cost function (4) may yield solutions which violate the exact fixed-point nature of the memories. How severe is this violation? This should be quantified and discussed.
Line 254-256: The statements here were not clear to me. Please clarify.
The authors do not mention capacity issues. It would be nice to make some statement, even preliminary, about this important computational aspect.
It would be nice if the authors could discuss the computational benefits of Dale’s law. Namely, can this law be shown to enhance computational power under specific constraints? This is not necessarily the case, and may be a result of evolutionary and/or biophysical constraints, but it would be nice to relate to this.
Summary: A clear and well written paper tackling a classic problem in neural computation using novel control-theoretic tools, and demonstrating how a biologically plausible associative memory may function.
Author Feedback
Author rebuttal: We thank all reviewers for their time and thorough evaluation of our paper.

Reviewer 36

1. The use of \gamma=0.04 follows Ahmadian et al (2013), but has virtually no importance. Eq. 2 shows that \gamma can be set to anything, provided the synaptic weights are rescaled. If our "potential variables" u are interpreted as membrane potentials, \gamma=0.04 is such that g(u) spans a few tens of Hz when u spans a few tens of mV, which is roughly what is observed in V1 (cf. eg. Ferster's papers on cat V1). We will add a short note about this.

2. Our choice of a log-normal distribution of firing rates is intended to roughly capture the skewed and heavy-tailed nature of firing rate distributions observed in vivo (reviewed eg. in Roxin et al, J Neurosci 2011). We will add a reference to it.

3. As you correctly point out, the minimization of the cost function may yield solutions in which the memories are not exactly fixed points, but only slow points. We found that for each such slow point, the basin of attraction was large enough that a true fixed-point in fact existed that was very close to the desired memory pattern. We believe this robustness is already quantified in our analysis of the recall performance, since we consider a recall attempt successful if the network state reaches the intended pattern by some small margin (e.g., a recall trial would be unsuccessful if the final state deviated from the intended firing rate by as little as 3 Hz in only one of 100 neurons). We will clarify this in the text.

4. Two reviewers noticed some imprecision in lines 254-256. We acknowledge that comparing our networks with feedforward perceptrons with E-I inputs (ref [12]) was too much of a stretch, and generated confusion. In particular Figure 3c could give rise to ambiguous interpretations. We will rewrite this part into a more neutral description of the findings, deferring a more careful comparison to future work.

5. Memory capacity. We generically observed a tradeoff between the number of memory patterns in the cost function and the robustness of their stability, as quantified by the corresponding values of both the Jacobian SA and the norm of the velocity vector. Figure 1a and later figures (which all correspond to m=30, cf our clarification below) show that the capacity is at least 0.2, but we later revisited the case m=50 and found all attractors to be stable, implying a capacity of at least 0.33. We will add these preliminary statements, together with statements about statistical significance to the main text (cf our answer to Rev. 8 below).

6. Concerning Dale's Law: we have no good reason to suspect that adding a constraint in our cost function would improve its minimization -- on the contrary, it may even make it more difficult. We will add this note to our discussion.

Reviewer 7

1. Our approach of building networks that embed arbitrary analog memory patterns inherently requires stability optimization. Few methods have been described in the literature of robust control to do just that. These methods correspond to different relaxations of the spectral abscissa (which is hopelessly ill-behaved, though some complex nonsmooth optimization procedures do exist, cf e.g. spectral bundle methods [Noll and Apkarian, Math. Progr. 2005]). For example, one could try and minimize the pseudo-spectral abscissa of the Jacobian [Burke et al., SIAM J. Matrix Anal Appl, 2003], with guarantees in terms of distance from instability (robustness). However the SSA minimization we use here is by far the simplest method to implement together with further constraints on the synaptic weights (Dale's law), and provides similar guarantees on robustness. Space limitations do not allow an in-depth discussion of those technical aspects, but we will provide pointers to those alternative methods.

2. Memory capacity: see point 5 for Rev. 36 above.

3. Expected impact: to our knowledge, our work is not simply "incremental", because it is the first example of control-theoretic techniques applied in this context, and because the resulting neural networks have the crucial combination of cortex-like capabilities and features that previous work simply failed to provide.

Reviewer 8

1. The velocity term is a crucial term of the cost function, because a stable Jacobian does by no means imply the fixed point condition (zero velocity). The Jacobian originates from a first order approximation, which is a good (local) approximation of the system only if the zeroth-order term (velocity at the point of interest) is negligible. Optimizing the Jacobian SA alone can easily make the velocity term grow and ultimately dominate. We will add a note about this before eq. 4.

2. Cueing. As in the Hopfield model, on test trials we set the initial neural activities in the proximity of an attractor, and we let the system evolve spontaneously towards a fixed point of its dynamics. Following initialization, the only external inputs that remain are the constant currents, {h_i} in Eq. 2, which are the same for all attractors. See also point 3 in our answer to Rev. 36. We will clarify this in the text.

3. We will add full details on the machine used to perform our computations.

4. Network trained with 50 attractors: see point 5 made for Rev. 36 above.

5. Statistical validation. At the time of our submission, we had produced a limited number of instances of the 30-attractors network and the 50-attractors network, plus various test-models of smaller sizes. Since then, we have ran many more instances, and our findings have always been consistent with those in the paper. We will refer to these more extensive simulations in the paper.

6. We will improve the text, correct the typos and change the labels of Figure 3d as suggested - we thank the reviewer for these.

7. Impact: see point 3 in answer to rev. 7.